# Learning Invariant Reward Functions through Trajectory Interventions

## Abstract

Inverse reinforcement learning methods aim to retrieve the reward function of a Markov decision process based on a data set of expert demonstrations. The commonplace scarcity of such demonstrations potentially leads to the absorption of spurious correlations in the data by the learning model. Consequently, this adaptation often exhibits behavioural overfitting to the expert data set when trained on the obtained reward function. The invariant risk minimization principle provides us with a novel regularization approach for the maximum entropy inverse reinforcement learning problem. By applying this regularization to both exact and approximate formulations of the learning task, we recover superior reward functions across domains in the transfer setting and, as a result, induce better performing policies than the empirical risk minimization baseline.

## 1 Introduction

Recent advancements in reinforcement learning methods applied to real world scenarios have revealed a major limitation, namely how to design a problem specific reward function. This design challenge typically requires a cumbersome and error-prone process of handcrafting a heuristic function to account for all the intricacies of the task at hand. Eliciting the correct behaviour via the optimization of a reward function is of paramount importance both for training artificial agents, such as robots, as well as human agents, for instance, in the case of teaching skills in a simulated environment, but modelling rewards for behavior learning remains an open challenge in a large number of domains.

Inverse reinforcement learning (IRL) solves the problem of inferring the reward function of a Markov Decision Process (MDP) based on a dataset of temporal behaviors. Such trajectories obtained from an agent are assumed to demonstrate near-optimal performance in the respective MDP. The problem to recover the reward from the statistics of the expert trajectories is ill-posed, as there typically exist many reward functions which satisfy the optimization constraints (Ng et al., 1999). Maximum entropy inverse reinforcement learning resolves this ambiguity in a unique way. Typically, the amount of expert demonstrations used to infer the reward is limited, which poses the danger of overfitting to the noise in the expert demonstrations. In particular, high-dimensional, parameterized reward models without appropriate regularization tend to absorb spurious correlations present in the trajectory data which do not generalize outside of the training setup. While optimizing the empirical cumulative reward obtained by such models, such an adaptation causes the agent to learn behaviours which frequently fail at test time. This type of observational overfitting has been shown in the context of reinforcement learning (Song et al., 2019). An additional difficulty arises when there is a significant discrepancy between expert demonstrations originating from different experts. Adversarial imitation methods fail to recover accurate behaviour from diverse experts as shown in (Li et al., 2017), since such learning results in ambiguities when inferring reward functions. We require to learn reward functions that elicit *succinct behaviours* of agents. Such rewards should avoid adopting behavioural features of experts that might be irrelevant to the task or even detrimental as a solution strategy.

Invariant risk minimization (IRM) (Arjovsky et al., 2019) exploits the concept of invariant causal prediction (Peters et al., 2015; Heinze-Deml et al., 2017). This recently introduced approach studies the generalization problem of classification models through the lens of causality. The IRM method postulates that the conditional distribution of the class label must be stable across datasets, in order to avoid absorption of spurious correlations into the model predictions. This adaptation principle can be utilized in the context of reward function learning, where we aim to elicit behavioural policies

without exploiting spurious reward features, i.e. features which cause nontransferable behaviours. To achieve this goal, we make the assumption that variations in expert demonstrations are a product of causal interventions on the data generating process of trajectories. Our contributions are as follows:

- We formulate the assumption that the variations between experts performing optimally on the same task can be seen as interventional settings of the underlying trajectory distribution.
- We propose a regularization principle based on invariant risk minimization. This modelling choice allows us to learn reward functions that are invariant to spurious correlations present in the expert data.
- We demonstrate the efficacy of this approach in both tractable, finite state-spaces, which we refer to as the exact setting as well as large continuous state-spaces, which we denote as approximate setting.

The rest of the paper is structured as follows: we provide an overview of related work in Section 2 and introduce the problem setting from the perspective of IRL as well as causal considerations in Section 3. We then present our model for the exact and approximate settings in Section 4 and demonstrate the experimental results in Section 5. We conclude with a brief discussion and outlook in Section 6.

## 2 RELATED WORK

**IRL and imitation learning**   Maximum entropy inverse reinforcement learning has been introduced in (Ziebart et al., 2008). An extension to functions parameterized by deep neural networks has been presented in (Wulfmeier et al., 2015). A formulation which considers large continuous state spaces where an approximation of the partition functions is required has initially been studied in (Finn et al., 2016) and expanded to an imitation setting in the work by (Ho & Ermon, 2016). A disentangled version of the adversarial imitation approach which allows to recover rewards has been presented in (Fu et al., 2017). The authors of (Zolna et al., 2019) propose a model which also tackles the issue of spurious correlations being absorbed from expert data. However, their focus is on visual features in a solely imitation learning setting as opposed to our approach, which recovers reward functions that perform favorably in a transfer setting.

**Invariant representation learning**   Invariant risk minimization is a method proposed in (Arjovsky et al., 2019) which builds upon the invariant causal prediction (Peters et al., 2015; Heinze-Deml et al., 2017) principle and proposes a way to tackle the issue of spurious correlations impeding out-of-distribution generalization of classification algorithms. Various other formulations of the IRM principle have subsequently been proposed. IRM games (Ahuja et al., 2020) reformulates the IRM objective from a game theoretic perspective and achieves lower variance compared to the original IRM method. Invariant rationalization proposes an adversarial method for recovering invariant rationales in the context of natural language processing (Chang et al., 2020) by employing an auxiliary model which is trained to be agnostic to interventional settings.

**Invariance and causality in RL**   The concept of invariance has been used in a number of works in the reinforcement learning domain. Invariant causal prediction has been utilized in (Zhang et al., 2020) to learn model invariant state abstractions in a multiple MDP setting with a shared latent space. Invariant policy optimization (Sonar et al., 2021) uses the IRM games (Ahuja et al., 2020) formulation to learn policies invariant to certain domain variations. The authors of (de Haan et al., 2019) tackle the problem of causal confusion in imitation learning by making use of causal structure of demonstrations. (Angelov et al., 2020) address the problem of user specification in robotic learning through the causal lens by augmenting the images with specific symbols. To the best of our knowledge, our algorithm is the first proposed method to use invariant causal prediction in the context of inverse reinforcement learning.

## 3 PROBLEM SETTING

**MDP**   We consider environments modelled by a *Markov decision process* $\mathcal{M} = (\mathcal{S}, \mathcal{A}, \mathcal{T}, P_0, R)$, where $\mathcal{S}$ is the state space, $\mathcal{A}$ is the action space, $\mathcal{T}$ is the family of transition distributions on $\mathcal{S}$ indexed by $\mathcal{S} \times \mathcal{A}$ with $p(s'|s, a)$ describing the probability of transitioning to state $s'$ when taking

action $a$ in state $s$, $P_0$ is the initial state distribution, and $R : \mathcal{S} \times \mathcal{A} \to \mathbb{R}$ is the reward function. A *policy* $\pi$ is a conditional probability distribution of actions $a \in \mathcal{A}$ given states $s \in \mathcal{S}$ with state descriptors or features $\varphi(s)$.

**Inverse reinforcement learning** aims to estimate a suitable reward function $r_\psi$ parameterized by weights $\psi$ based on a dataset of expert trajectories $\mathcal{D}_E = \{\tau_i\}_{i \leq K}$ where $\tau_i = (s_{1:T}^{(i)}, a_{1:T}^{(i)})$ is a sequence of states and actions of expert $i$ of length $T$. To achieve this goal, the feature expectation statistics under the state occupancy measure of the expert, $\mathbb{E}_{\tau \sim \mathcal{D}_E}[\varphi(\tau)]$, are matched with the statistics of the student state occupancy measure $\mathbb{E}_{\tau \sim p(\tau|\psi)}[\varphi(\tau)]$. $\varphi(\tau) = \sum_{s \in \tau} \varphi(s)$ denotes the sum of the state features $\varphi(s)$. The trajectory distribution $p(\tau|\psi)$ is induced by the policy $\pi_{r_\psi}$ trained on the reward function estimate $r_\psi$. Given the ill-posed nature of the problem, a maximum entropy formulation of the IRL problem (Ziebart et al., 2008) is typically chosen in order to find a unique solution. The model describes the fact that the expert trajectories are sampled from the Gibbs distribution $p(\tau|\psi, \varphi) = \frac{1}{Z_{\varphi,\psi}} \exp(r_\psi(\tau)) = \frac{1}{Z_{\varphi,\psi}} \exp(\psi^T \varphi(\tau))$ which corresponds to the solution of the entropy maximization problem under feature matching and simplex constraints.

$$\max_\psi H(p(\tau|\psi)) \text{ s.t. } \left\{ \mathbb{E}_{\tau \sim p(\tau|\psi)}[\varphi(\tau)] = \mathbb{E}_{\tau \sim \mathcal{D}_E}[\varphi(\tau)]; \int_\mathcal{T} p(\tau|\psi)d\tau = 1; \ p(\tau|\psi) > 0 \right\} \quad (1)$$

where $\mathbb{E}_{\tau \sim \mathcal{D}_E}[\varphi(\tau)]$ is the sample average of the expert trajectories. In the feature matching case, the reward $r_\psi(\tau)$ is typically assumed to be linear in the state features $r_\psi = \psi^T \varphi(\tau)$. We further consider the scenario where the state features $\varphi(\tau)$ are learned using the DeepMaxEnt model (Wulfmeier et al., 2015). In large state spaces, the partition function $Z$ presents an estimation challenge and an importance sampling scheme is typically employed.

**Adversarial IRL formulation**   It has been shown (Finn et al., 2016) that the trajectory distribution for the importance sampling scheme can be obtained by solving an adversarial minimax game implemented by a generative adversarial network (Goodfellow et al., 2014) architecture. Adversarial inverse RL (AIRL) methods yield a reward function by learning to distinguish between the transitions sampled from the dataset of expert trajectories $\tau \sim \mathcal{D}_E$ and transitions continually sampled from the improving policy $\tau \sim \pi(\tau)$, which maximizes an expected cumulative reward objective based on the discriminator output. We adopt the discriminator structure from (Fu et al., 2017) which separates into the state-action dependent reward function $g_\xi$ and the state dependent shaping term $h_\phi$.

$$D_{\xi,\phi,\theta}(s, a, s') = \frac{\exp(g_\xi(s, a) + \gamma h_\phi(s') - h_\phi(s))}{\exp(g_\xi(s, a) + \gamma h_\phi(s') - h_\phi(s)) + \pi_\theta(a|s)} \quad (2)$$

We also consider a state-only formulation of AIRL where the reward function $g_\xi(s)$ only depends on the state $s$. The optimization objective of the discriminator is the binary cross-entropy loss on the discriminator output:

$$\mathcal{L}_{\text{BCE}}(\xi, \phi, \theta) = \mathbb{E}_{(s,a) \sim \pi_\theta} \log(D_{\xi,\phi,\theta}(s, a)) - \mathbb{E}_{(s,a) \sim \pi_E} \log(1 - D_{\xi,\phi,\theta}(s, a)) \quad (3)$$

The generator corresponds to the policy optimizing the expected cumulative reward given by the discriminator output and is trained using the proximal policy optimization (PPO) (Schulman et al., 2017) method.

As with many neural network approximators, the discriminator model absorbs spurious correlations and this learning effect poses a serious problem (Arjovsky et al., 2019). These correlations coincide with the binary label information encoding the optimality of the expert. In this work, we avoid this effect by applying the invariant risk minimization principle to the discriminator part of our model.

**Invariant risk minimization** (Arjovsky et al., 2019) extends the invariant causal prediction (Peters et al., 2015; Heinze-Deml et al., 2017) principle to nonlinear settings where the underlying SCM might not be explicitly given. The goal of IRM is to learn correlations which are invariant across training *settings* and ignore spurious correlations. The set of training settings $\mathcal{E}_{tr}$ contains datasets $D_e :=$

$\{(x_e^{(i)}, y_e^{(i)})\}_{i=1}^{|\mathcal{E}_{tr}|}$ sampled from interventional distributions $\tilde{P}(X_e, Y_e)$ of the SCM of the underlying data generating process. The IRM method effectively learns an invariant data representation $\varphi$ for the predictor $w \circ \varphi$ (where $w$ is a linear classifier) which enables a stable conditional distribution $P(Y^e | X^e)$ by optimizing the objective stated in the definition below:

**Definition 1** *The data representation $\varphi : \mathcal{X} \to \mathcal{H}$ mapping the input to a hypothesis space, elicits an invariant predictor $w \circ \varphi$ across the set of interventional settings $\mathcal{E}$ if there is a classifier $w : \mathcal{H} \to \mathcal{Y}$ simultaneously optimal for all settings:*

$$\min_{\varphi:\mathcal{X}\to\mathcal{H}, w:\mathcal{H}\to\mathcal{Y}} \sum_{e\in\mathcal{E}_{tr}} \mathcal{L}^e(w \circ \varphi) \quad \text{s. t.:} \quad w \in \operatorname*{argmin}_{\bar{w}:\mathcal{H}\to\mathcal{Y}} \mathcal{L}^e(\bar{w} \circ \varphi) \quad \forall e \in \mathcal{E}_{tr} \tag{4}$$

The aim of this bi-level optimization problem is to minimize the prediction loss on the union of the training sets constrained on selecting the predictor which minimizes the losses on the individual training set e, $\mathcal{L}^e$. The tractable approximation of Eq. 4 uses the gradient norm penalty $\mathbb{D}(w = 1.0, \varphi, e) = ||\nabla_{w|w=1.0}\mathcal{L}^e(w \circ \varphi)||^2$ which quantifies the violation of the normal equations to measure the optimality of a fixed classifier ($w = 1.0$) at each setting $e$. This leads to the following unconstrained formulation of the problem:

$$\min_{\varphi:\mathcal{X}\to\mathcal{Y}} \sum_{e\in\mathcal{E}_{tr}} \mathcal{L}^e(\varphi) + \lambda ||\nabla_{w|w=1.0}\mathcal{L}^e(w \circ \varphi)||^2 \tag{5}$$

In the following section, we will show how to incorporate this optimization principle into the IRL setting.

## 4 MODEL

In this section, we present the invariance regularization for the maximum entropy IRL models in exact and approximate settings which allows us to eliminate spurious correlations present in the expert datasets. We assume that different experts stem from interventional settings of the trajectory SCM. We first present the types of interventions on the trajectory SCM that we consider in this work. In the second subsection, we show how to apply this principle in a feature matching scenario. In the final section, we propose a regularization method for the adversarial learning scenario, which penalizes the discriminator function in order to exclude spurious correlations when distinguishing between policy and expert transitions.

### 4.1 INTERVENTIONS

The principle of invariant risk minimization relies on the assumption that the difference in training settings arises as a result of an intervention on the data generating process described by a structural causal model. In the case of maximum entropy IRL, the generative model is the SCM of trajectories $p(\tau|\mathcal{O}_{1:T})$ conditioned on the optimality variable $\mathcal{O}_{1:T}$ for timesteps $t = \{1, ..., T\}$ factorizes into the initial state distribution $p_0(\mathbf{s}_1)$, the conditional distribution $p(\mathcal{O}_t|\mathbf{s}_t, \mathbf{a}_t) = \exp(r_\psi(\mathbf{s}_t, \mathbf{a}_t))$ of the binary optimality variable $\mathcal{O}_t$ at timestep $t$ and the MDP transition dynamics $p(\mathbf{s}_{t+1}|\mathbf{s}_t, \mathbf{a}_t)$ (Levine, 2018) is defined by the following expression:

$$p(\tau|\mathcal{O}_{1:T}) \propto p(\tau, \mathcal{O}_{1:T}) = p_0(\mathbf{s}_1) \prod_{t=1}^{T} p(\mathcal{O}_t = 1|\mathbf{s}_t, \mathbf{a}_t)p(\mathbf{s}_{t+1}|\mathbf{s}_t, \mathbf{a}_t) \tag{6}$$

**Assumption 1** *Different expert demonstrations stem from interventional settings of the trajectory SCM.*

The assumption is valid in a scenario where expert demonstrations were gathered on different dynamics. For instance, recordings of surgeons performing a procedure on different patients with variations of anatomy would satisfy this assumption.

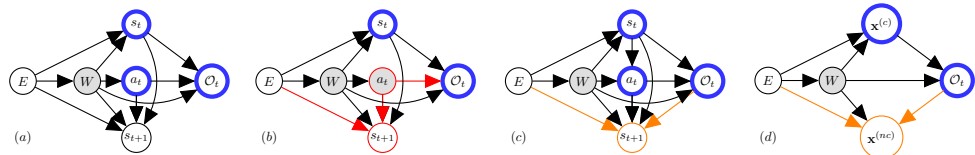

Figure 1: (a) Probabilistic graphical model of a transition under influence of the index variable $E$ and latent variable $W$. The stable conditional is highlighted in blue. (b) Spurious correlations assuming state-only formulation. (c) Spurious correlations assuming wrong edge orientation $\mathcal{O}_t \rightarrow s_{t+1}$. (d) General setting where $\mathcal{O}_t$ depends on causal $\mathbf{x}^{(c)}$ and non-causal $\mathbf{x}^{(nc)}$ features of the transition.

In order to apply the IRM principle to the inverse reinforcement learning problem, we must obtain a set of settings $\mathcal{E}$ which all contain the relation of interest invariant with respect to these settings. For the trajectory model state above, the causal relation of interest in the context of reward function inference is the conditional $P(\mathcal{O}_t|s_t, a_t)$, which should be invariant under interventions to the transition SCM (Fig. 1a). We motivate this by the fact that despite the discrepancies in the demonstrations, all experts are assumed to perform the task in an optimal fashion. This implies that all experts optimize the same underlying reward that we would like to recover. By considering the conditional independence structure in the graphical model of the state-action transition in (Fig. 1a) we can observe the underlying assumption that the causal mechanism producing the optimality label $\mathcal{O}_t$ is invariant across variations of the setting index variable $E$ provided that the latent variable $W$ remains unobserved.

**Spurious correlations in model of transitions** We will now describe the two scenarios in which a non-causal information path corresponding to spurious correlations is formed in the transition SCM. In Fig. 1b we can observe the scenario where we do not condition our reward function reprentation $r(s)$ on the action. By conditioning on the collider node $s_{t+1}$ and not observing the action node $a$, a path is formed between the setting index $E$ and the optimality variable $\mathcal{O}_t$, resulting in the violation of their conditional independence relationship. A second scenario can be observed in Fig. 1c. This scenario requires the assumption that the orientation of the edge from node $\mathcal{O}_t$ to node $s_{t+1}$ is temporally causal, meaning that the optimality of a state at time $t$ is a causal parent of the next state. In this case, observing the collider node $s_{t+1}$ implies the following conditional independence relationship: $E \not\perp O_t|s_{t+1}$. Beyond these scenarios, one can further assume a more general partitioning of an arbitratry transition input $(s, a, s')$ into the causal transition feature components $\mathbf{x}^{(c)}$ and $\mathbf{x}^{(nc)}$ : $(s, a, s') = (\mathbf{x}^{(c)}, \mathbf{x}^{(nc)})$, illustrated in Fig 1d, whereby conditioning on the $\mathbf{x}^{(nc)}$ collider introduces a spurious correlation path (Huszar, 2019).

The first scenario arises in the feature matching case where the trajectory feature representation $\varphi(\tau)$ learned using the DeepMaxEnt (Wulfmeier et al., 2015) model only has access to the subsequent state but not the action observations. The second scenario arises in the adversarial learning case where the discriminator $D_{\xi,\phi,\theta}(s, a, s')$ 2 depends on the node $s'$, hence observing a collider.

Following the assumption of the IRM principle that we only make interventions on non-causal parents of the optimality variable $\mathcal{O}_t$, possible interventions are state interventions caused by altering the initial state distribution or the dynamics of the MDP and the action interventions, caused by intervening on the agent policy.

## 4.2 MAXIMUM ENTROPY IRL REGULARIZATION

For the application of the IRM regularization, we first consider the maximum entropy feature matching scenario. For the Gibbs distribution $p(\tau|\psi, \varphi) = \exp(\psi^T\varphi(\tau))/Z_{\varphi,\psi}$ over trajectories, we can write down the constrained optimization problem analogously to Eq. 4:

$$\min_{\varphi,\psi} \sum_{e \in \mathcal{E}_{tr}} \sum_{\tau \in \mathcal{D}_e} \log p(\tau|\psi, \varphi) \text{ s.t. } \psi \in \operatorname*{argmin}_{\bar{\psi}} \sum_{\tau \in \mathcal{D}_e} \log p(\tau|\bar{\psi}, \varphi) \qquad (7)$$

In simple settings such as gridworlds, where the computation of the partition function is tractable, we propose the following regularization approach for the MaxEnt maximum likelihood objective:

$$\min_{\varphi} \sum_{e \in \mathcal{E}_{tr}} \left( \sum_{\tau \in \mathcal{D}_e} \log \left( \frac{1}{Z_{\psi,\phi}} \exp(\psi^T \varphi(\tau)) \right) + \lambda \mathbb{D}(\psi, \varphi, e) \right) \tag{8}$$

In an analogous fashion to the invariant risk minimization approach, $\mathbb{D}(\psi, \varphi, e)$ is a distance function representing the violation of the constraints w.r.t. the optimal solution, which in our case corresponds to the parameters maximizing the likelihood of the Gibbs distribution. The gradient of the log-likelihood loss $\mathcal{L}_{MLE} = \sum_{\tau \in \mathcal{D}_e} \log p(\tau | \psi, \varphi)$ w.r.t. to $\psi$ is then the difference of the feature expectations $\mathbb{E}_{\mathcal{D}_E}[\varphi(\tau)] - \mathbb{E}_{p(\tau|\psi)}[\varphi(\tau)]$ (Ziebart et al., 2008). The squared norm of this gradient constitutes the equivalent of the IRM penalty in the maximum entropy IRL case.

$$\mathbb{D}(\psi, \varphi; e) = ||\nabla_{\psi|\psi=1.0} \mathcal{L}_{\text{MLE}}(\psi, \varphi)||^2 = ||\mathbb{E}_{\mathcal{D}_E}[\varphi(\tau)] - \mathbb{E}_{p(\tau|\psi)}[\varphi(\tau)]||^2 \tag{9}$$

This closed form of the gradient norm penalty can be utilized in the maximum causal entropy solver (Ziebart et al., 2010). We assume the state features to be the output of a neural network $\varphi_{\theta}(s)$ according to the DeepMaxEnt model (Wulfmeier et al., 2015). The gradient norm penalty 9 is then backpropagated to enforce invariant features $\varphi$.

### 4.3 ADVERSARIAL IRL REGULARIZATION

We now present the adversarial version of the invariant regularization in algorithm 1. The training procedure of alternating policy updates using a policy gradient method with discriminator updates is similar to (Fu et al., 2017). There are two main differences compared to the baseline algorithm. The first is the fact that we use multiple experts in a distinct fashion as opposed to pooling the demonstrations into one big dataset. The second is the regularization of the discriminator objective (Eq. 3) using the gradient norm penalty $\mathbb{D}(\xi, \phi, \omega; e) = ||\nabla_{\omega|\omega=1.0} \mathcal{L}_{\text{BCE}}(\xi, \phi, \omega; e)||^2$ in a similar fashion to Eq. 4, where $\omega = 1.0$ corresponds to a fixed scalar classifier. The regularized discriminator effectively iterates over tuples $(\mathcal{D}_E^e, \mathcal{D}_\pi) \; \forall e \in \mathcal{E}_{tr}$ during adversarial training, where $\mathcal{D}_\pi$ is dataset of transitions generated by policy $\pi$.

---

**Algorithm 1:** IRM regularized adversarial IRL

---

**Input:** Expert trajectories $\mathcal{D}_E^e$ assumed to be obtained *by intervening on* $p(\tau)$ in settings $e$;
**Result:** Reward $r_\xi$, trained student policy $\pi_\theta$
Initialize policy $\pi_\theta$ and discriminator $D_{\xi,\phi}$;
**for** *setting e in* $\{1, ..., \mathcal{E}_{tr}\}$ **do**
 **for** *step t in* $\{1, ..., N\}$ **do**
  Collect trajectory buffer $\mathcal{D}_\pi = \{\tau_i\}_{i \leq |\mathcal{D}_\pi|}$ by executing the policy $\pi_\theta$;
  Update $D_{\xi,\phi,\theta}(s, a)$ via binary logistic regression by maximizing;

$$\mathcal{L}(\xi, \phi, \omega; e) = \mathcal{L}_{\text{BCE}}(\xi, \phi, \omega; e) + \lambda ||\nabla_{\omega|\omega=1.0} \mathcal{L}_{\text{BCE}}(\xi, \phi, \omega; e)||^2$$

  using *expert-specific* tuple $(\mathcal{D}_E^e, \mathcal{D}_\pi)$
  Update policy $\pi_\theta$ w.r.t. $r_{\psi,\phi} = \log D_{\xi,\phi}(s, a, s') - \log(1 - D_{\xi,\phi})$ function of *IRM*
   *regularized discriminator* using a policy gradient method (e.g. PPO);
 **end**
**end**

---

The training process yields two outputs: the trained reward component $r_\xi$ of the discriminator model as well as the trained student policy. In our experiments we show that the policy obtained through the adversarial training procedure shows improved zero-shot generalization performance on environments with dynamics sampled outside of the training distribution compared to the baseline. The performance is measured using the ground truth reward. Furthermore, we show that rewards obtained using this procedure elicit better performing policies when trained from random initialization on environments with modified dynamics.

## 5 EXPERIMENTS

The experiments are designed to answer the following questions:

- Can the MaxEnt IRL setting benefit from applying the invariance principle?
- Is the IRM principle helpful for adversarial reward learning to infer invariant discriminators?
- Do causal assumptions improve robustness of reward functions?

To answer these questions, we evaluate our model in three settings. The first setting considers a grid-world scenario, where the partition function is tractable and the reward is recovered using our modification of the maximum entropy feature expectation matching algorithm. In the second setting, we test the invariance regularization in an adversarial setting on simulated robotic locomotion environments. Finally, we demonstrate generalization of the obtained reward functions by retraining policies on the reward functions obtained using the adversarial methods.

Throughout this section, we compare the empirical risk minimization (ERM) baseline where the trajectory datasets gathered through interventions are pooled together into one dataset to the IRM version of the algorithms where we regularize either the feature expectation matching algorithm (sec. 5.1) or the adversarial formulation by assuming interventions on the expert trajectories (sec. 5.2, 5.3) as described in section 4.

### 5.1 TRACTABLE SETTING: GRIDWORLD EXPERIMENTS USING FEATURE MATCHING

To illustrate the principle of invariant risk minimization in the tractable IRL setting as discussed in 4.2, we choose a simple gridworld environment illustrated in (Fig. 2a). The goal of the agent is to navigate from the bottom left to the top right corner. The gridworld has stochastic dynamics: the chance of uniformly transferring to a state around the target is $p_{slip} = 0.2$.

**Setup**  In order to construct dataset settings, we generate dataset of 3 groups of expert trajectories by training the policies using a value iteration method on modified versions of the MDP. The initial and final states of the trajectories are fixed. We introduce a selection bias into the IRL feature expectation matching problem by manipulating the trajectory dataset in a way that we have a different number of trajectories for each of the three paths chosen by the experts (Fig. 2a): 400 trajectories for 1st group, 25 trajectories for 2nd group, 3 trajectories for 3rd group.

**Results**  In Fig 2, we can observe that both the unregularized MaxEnt IRL algorithm (ERM) (Fig. 2b) and L2-regularized MaxEnt IRL algorithm (ERM) (Fig. 2c) exhibit overfitting to the expert datasets and partially to recover a meaningful reward and respective policy where as the IRM-regularized version recovers a shaped reward function which takes the different optimal paths into account in a correct manner. In particular, increasing the regularization strength $\lambda$ improves the reward significantly. (Fig. 2d - Fig. 2e).

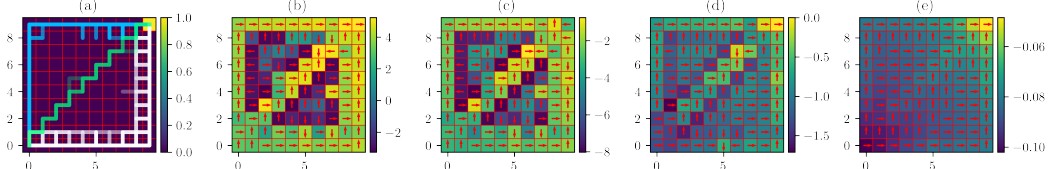

Figure 2: Gridworld ERM vs IRM reward recovery. (a) expert trajectory datasets: 1st group (blue) 400 trajectories, 2nd group (white): 25 trajectories, 3rd group (green): 3 trajectories. (b) MaxEnt IRL ERM baseline (c) MaxEnt IRL ERM baseline with L2 regularization coefficent 1e-3 (d) MaxEnt IRL with IRM penalty, $\lambda = 0.01$, (e) MaxEnt IRL with IRM penalty, $\lambda = 0.05$

### 5.2 ADVERSARIAL SETTING: PYBULLET ROBOT ENVIRONMENTS

**Setup**  In this section, the experiments are performed in PyBullet (Ellenberger, 2018–2019) gym (Brockman et al., 2016) environments, which are an open-source alternative to the MuJoCo physics

simulator. We generate the demonstration datasets by training policies using the PPO algorithm (Schulman et al., 2017) [1] and varying the dynamics of the environments. An environment is defined as an instance of the variable $E$, which has an impact on the prior distribution of transitions (and by extension, trajectories) $p(s, a, s')$ and $p(\tau)$ in a minibatch of rolled out on-policy and expert transitions. We have used 10 expert trajectories for every environment for the physical parameter modification experiments. For the goal intervention experiments, we have used a proportion of 1, 3 and 10 trajectories for the three goal directions respectively.

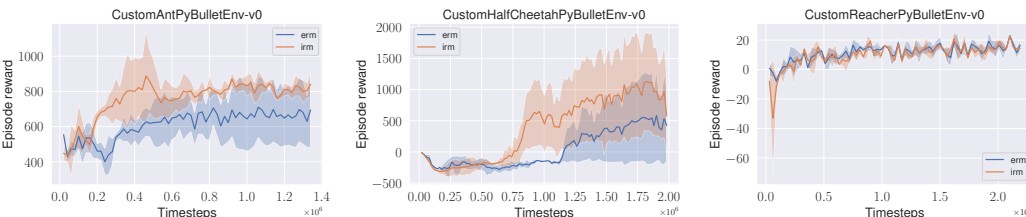

Figure 3: Policy performance of the ERM vs IRM methods on out-of-distribution settings in the physical parameter intervention setting

**Results on physical parameter interventions** We test the performance using the policy rollout method with respect to the ground truth reward. We compare the policies trained using the adversarial approach with and without the invariance penalty. The evaluation environment setting is chosen to be outside of the value range presented to the model at training time. We can observe in Fig. 3 that the IRM-regularized version of AIRL outperforms the baseline in a zero-shot generalization setting in both of the locomotion environments (`CustomAntMuJoCo` and `CustomHalfCheetahMuJoCo` [2]), where interventions were performed on the hind leg length (2x and 3x of original length) and friction coefficients respectively. The friction coefficients have been modified to have a value of $(1.5, 0.1, 0.1)$ and $(2.0, 0.6, 0.6)$ tangential, torsional and rolling coeffients respectively. In contrast, we can see that for the `CustomReacherPyBulletEnv` environment, where the demonstrations were recorded by policies acting on the environment with a varying gravity coefficient, both models perform similarly. We attribute this to the fact that the gravity plays little role in the context of the Reacher environment.

**Results on goal interventions** We have also performed a second set of experiments using the following trajectory modification strategy. By altering the target y-coordinate of the locomotion environments and effectively changing the goal direction, we have generated a set of three distinct trajectory distributions. We have further introduced a selection bias into the dataset by selecting a higher number of trajectories which favors the right ($y_{target} = -900$) in addition to the trajectories with the default goal position. The target y-coordinate in the out-of-distribution testing environment has been chosen to be the hard left ($y_{target} = 900$). We can see an even stronger improvement in performance in IRM-regularized models (fig. 4) when compared to the ERM baseline, which fails to learn an effective policy in the case of the Half Cheetah.

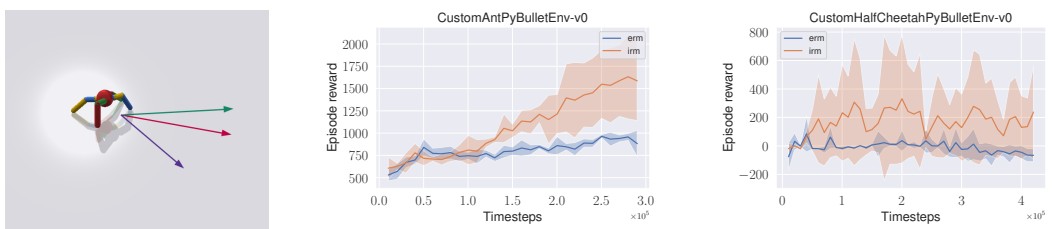

Figure 4: Policy performance of the ERM vs IRM methods on out-of-distribution environments in a goal intervention setting

---

[1] stable-baselines3 implementation

[2] *MuJoCo refers to the environment (MDP) specification. The simulation engine is PyBullet.

Table 1: Transfer performance on out-of-distribution (OOD) environments. Standard deviation values are computed over different random seeds of the same OOD environment. The statistical significance p-values are 0.0458, 0.0456, 0.0041 and 0.0052 respectively for the four scenarios below ($p \leq 0.05$).

| Environment | Model | | |
| --- | --- | --- | --- |
| | AIRL-ERM | AIRL-IRM | Expert |
| LunarLander-v2 (bounce) | $93.1_{\pm 30.6}$ | $156.8_{\pm 53.1}$ | $230.3_{\pm 15.6}$ |
| LunarLander-v2 (engine power) | $54.3_{\pm 64.5}$ | $134.7_{\pm 40.2}$ | $204.8_{\pm 22.3}$ |
| CustomAntMuJoCo-v0 (friction) | $542.7_{\pm 114.5}$ | $783.9_{\pm 72.4}$ | $1223.2_{\pm 187.2}$ |
| CustomAntMuJoCo-v0 (leg length) | $640.7_{\pm 87.3}$ | $859.3_{\pm 94.1}$ | $1394.6_{\pm 154.3}$ |

### 5.3 REWARD TRANSFER

In this experimental scenario, we use trained reward functions extracted from the discriminator model in order to retrain a new policy from scratch. The training dynamics of the policy with respect to the frozen version of the reward parameters is different as opposed to the training dynamics during adversarial training due to the varying discriminator quality during training. In order to train the models, we use the same reference PPO implementation as for the generation of the expert trajectories in the previous setting. In this scenario, we additionally use the `LunarLander-v2` environment as a benchmark with interventions on the bounce parameter for the interaction of the landing gear and the platform as well as the engine power multiplier.

In table 1, we can observe that the policies trained on the rewards obtained through regularized adversarial training show improved performance compared to the ERM baseline in terms of cumulative reward but fail to reach expert performance which corresponds to a policy trained using the ground truth reward. We consider this to be of limited importance since we typically do not have access to the ground truth reward function in an IRL setting.

## 6 CONCLUSION

We have presented a regularization objective for inverse reinforcement learning to recover robust reward functions which avoid to learn spurious correlations present in demonstration data sets. The robustness manifests itself as improved generalization performance in out-of-distribution settings both in the maximum entropy IRL case based on feature expectation matching as well as the adversarial setting. The student policies that optimize the adversarial discriminator signal regularized by our method, demonstrate improved zero-shot generalization capabilities as shown by our experiments.

An open challenge remains the question how to construct effective interventional settings, i.e., how to determine whether the interventional assumptions are fulfilled by the given expert datasets. While the presented model successfully improves upon the baseline, it remains an open question what is the correct degree of invariance the model should aim for while avoiding degenerate invariant solutions.

In future work, one could envision a setting where such interventions could be discovered in an adversarial manner, for instance, by leveraging procedurally generated environments.

In the process of adversarial training, the distribution over transitions $P(s, a, s')$ is subject to shifts due to an improving student policy $\pi_\theta$. The invariant regularization objective imposed on the discriminator and the exploitation of this effect by the student policy coupled with the adversarial training dynamics requires further analysis. Explicit use of the regularization term as a reward component for the student policy could be another avenue to explore.

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

## A    DERIVATION OF THE PENALTY TERM FOR MAXIMUM ENTROPY IRL

The gradient of the log-likelihood loss $\mathcal{L}_{MLE} = \sum_{\tau \in \mathcal{D}_e} \log p(\tau|\psi, \varphi)$ w.r.t. to $\psi$ is computed as follows:

$$\mathcal{L}_{MLE} = \sum_{\tau \in \mathcal{D}_e} \log p(\tau|\psi, \varphi) = \sum_{\tau \in \mathcal{D}_e} \log(\exp(\psi^T \varphi(\tau))) - \log Z_{\psi,\varphi} = \sum_{\tau \in \mathcal{D}_e} \psi^T \varphi(\tau) - \log Z_{\psi,\varphi}$$

Differentiating w.r.t. $\psi$ yields:

$$\frac{\partial L_{MLE}}{\partial \psi} = \mathbb{E}_{\mathcal{D}_e}[\varphi(\tau)] - \frac{1}{Z_{\psi,\varphi}} \int \exp(\psi^T \varphi(\tau))\varphi(\tau)d\tau$$

$$= \mathbb{E}_{\mathcal{D}_E}[\varphi(\tau)] - \mathbb{E}_{p(\tau|\psi)}[\varphi(\tau)]$$

The gradient penalty term from Eq. 9 with respect to the features $\varphi$ is derived as follows:

$$\nabla_\varphi \left\| \nabla_{\psi|\psi=1.0} L^e \left( r\left(\psi, \varphi\right)\right) \right\|^2 = \frac{\partial \| \frac{\partial L^e(r(\psi,\varphi)\cdot)}{\partial \psi}|_{\psi=1.0} \|^2}{\partial \varphi}$$

We employ the chain rule:

$$\frac{\partial L^e \left( r\left(\psi, \varphi\right)\right)}{\partial \psi} = \frac{\partial L^e \left( r\left(\psi, \varphi\right)\right)}{\partial r} \cdot \frac{\partial \left( r\left(\psi, \varphi\right)\right)}{\partial \psi} = \frac{\partial L^e \left( r\left(\psi, \varphi\right)\right)}{\partial r} \cdot \varphi$$

where the last equality holds because we assume a linear reward with respect to the features $\varphi$: $r\left(\psi, \varphi\right) = \psi^T \varphi$. Also, in section 4.2 we showed that :

$$\frac{\partial L^e \left( r\left(\psi, \varphi\right)\right)}{\partial r} = \mathbb{E}_{\mathcal{D}_E}[\varphi(\tau)] - \mathbb{E}_{p(\tau|\psi)}[\varphi(\tau)]$$

where $\mathbb{E}_{\mathcal{D}_E}[\varphi(\tau)]$ are the feature statistics of the expert and $\mathbb{E}[\rho_\pi]$ are the feature statistics of the student policy. For the sake of simplicity we define:

$C := \mathbb{E}_{\mathcal{D}_E}[\varphi(\tau)] - \mathbb{E}_{p(\tau|\psi)}[\varphi(\tau))$ which is independent of $\varphi$: Then:

$$\frac{\partial L^e \left( r\left(\psi, \varphi\right)\right)}{\partial \psi} = C\varphi$$

We obtain the IRM penalty for the feature matching maximum entropy IRL case as the following term:

$$\nabla_\varphi \left\| \nabla_{\psi|\psi=1.0} L^e \left( r\left(\psi, \varphi\right)\right) \right\|^2 = \frac{\partial \| \frac{\partial L^e(r(\psi,\varphi)\cdot)}{\partial \psi}|_{\psi|\psi=1.0} \|^2}{\partial \varphi} = \frac{\partial \left\| C\varphi \right\|^2}{\partial \varphi}$$

$$= \frac{\partial \left[ C\varphi \right]^T \left[ C\varphi \right]}{\partial \varphi}$$

$$= C^T C \frac{\partial \varphi^T \varphi}{\partial \varphi} = 2 \left\| C \right\|^2 \varphi = 2 || \rho_E - \mathbb{E}\left[ \rho_\psi \right] ||^2 \varphi$$

## B    STRUCTURAL CAUSAL MODELS

A structural causal model (SCM) is defined as a tuple $\mathfrak{G} = (S, P(\varepsilon))$, where $P(\varepsilon) = \prod_{i \leq K} P(\varepsilon_i)$ is a product distribution over exogenous latent variables $\varepsilon_i$ and $S = \{f_1, ..., f_K\}$ is a set of structural mechanisms where pa($i$) denotes the set of parent nodes of variable $x_i$:

$$x_i := f_i(\text{pa}(x_i), \varepsilon_i) \qquad \text{for } i \in |S| \tag{10}$$

$\mathfrak{G}$ induces a directed acyclic graph (DAG) over the variables nodes $x_i$ and entails a joint observational distribution $P_{\mathfrak{G}} = \prod_{i \leq K} p(x_i|\text{pa}(x_i))$ over variables $x_i$ conditioned on the parents of $x_i$ for some probability distribution $p(\cdot|\text{pa}(x_i))$ describing the mechanism $f_i$. Interventions on $\mathfrak{G}$ constitute modifications of one or more structural mechanisms $f_i$ yielding interventional distributions $\tilde{P}_{\mathfrak{G}}$. In this paper, we have considered interventional distributions of expert trajectories $\tilde{P}_{\mathfrak{G}_e}(\tau)$.

## C    MODEL ARCHITECTURE AND TRAINING DETAILS

For the gridworld experiments, we use a DeepMaxEnt (Wulfmeier et al., 2015) formulation of the IRL problem. The state features are parametrized by a 2-layer MLP with a 1-dimensional hidden layer. We use RMSProp as an optimizer with a learning rate of $1e - 3$.

For the adversarial learning experiments, we use an actor and critic network with two layers of 64 neurons ( `MlpPolicy` from stable-baselines3 (Raffin et al., 2019)) We use Adam (Kingma & Ba, 2014) as the optimizer with an initial learning rate of $\eta = 3e - 4$ and an L2 weight decay coefficient of $\lambda = 1e - 4$. We perform one update of the discriminator network for every 3 updates of the actor-critic networks. We use the tuned hyperparameters from `rl-baselines-zoo-3`[3] to train the policy networks both using the ground truth reward as well as during adversarial training and using the recovered reward for the transfer experiments.

## D    INTERVENTIONAL SETTINGS

Table 2: Interventional settings for the experimental environments (ID: in distribution, OOD: out-of-distribution)

| Environment | Intervention | Values (ID) | Values (OOD) |
|---|---|---|---|
| LunarLander-v2 | Bounce | 0.1-0.5 | 0.9 |
| | Friction | 0.2-0.4 | 0.001 |
| CustomAntMuJoCo-v0 | Mass | 0.5x, 2x | 4x |
| | Friction | (1.5,0.1,0.1), (2.0,0.6,0.6) | (2.5,0.7,0.7) |
| | Hind Leg Length | 1x,2x | 3x |
| | Direction angle | $\mathcal{U}(-10,0)$ | 10 |
| CustomHalfCheetahMuJoCo-v0 | Mass | 0.5x, 2x | 4x |
| | Friction | (1.5,0.1,0.1), (2.0,0.6,0.6) | (2.5,0.7,0.7) |
| | Leg Length | 0.7-1.3 | 0.5, 1.5 |
| | Direction angle | $\mathcal{U}(-10,0)$ | 10 |
| CustomReacherPyBulletEnv-v0 | Gravity | 0.5x,2x | 4x |

---

[3]https://github.com/DLR-RM/rl-baselines3-zoo

