# OpenReview forum: "Learning Invariant Reward Functions through Trajectory Interventions"
_ICLR.cc/2022/Conference — ICLR 2022 Submitted_

### Official Review · Reviewer_YvDw · 2021-10-31

**Correctness:** 3
**Technical Novelty And Significance:** 4
**Empirical Novelty And Significance:** 3
**Recommendation:** 8
**Confidence:** 3

**Main Review:**

Pros:
- The paper addresses a very important issue of IRL, i.e., the risk of overfitting the demonstrated trajectories. I think that the approach presented in the paper is very interesting and represents a starting point for a comprehensive treatment of the topic.
- The experimental evaluation, especially the gridworld experiment, effectively shows that the proposed approach is able to generalize very well, recovering a very smooth reward function. Moreover, the recovered rewards, in the LunarLander experiment, display better transferability properties compared to plain maximum entropy IRL.

Cons:
- The construction of the interventions, as the authors acknowledge, might be complex and the paper does not provide any suggestion on how to discover them
- Eq. (9) and Eq. (5) differ because in Eq. (9) there is a summation over the trajectories also in the constraint, while in Eq. (5) the constraint is stated for every intervention (universal quantification). Can the authors clarify?

Minor issues
- The ticks on the plot axis are very small
- Pag. 4: The definition of \mathcal{E}_{tr} is not clear. Is it a set of datasets, like \mathcal{E}_{tr} = \{D_{e_1},...,D_{e_m} \}?

**Summary Of The Paper:**

The paper studies the Inverse Reinforcement Learning (IRL) problem, addressing the issue of avoiding overfitting when having access to a finite set of expert demonstrations. The paper proposes an approach based on Invariant Risk Minimization (IRM) as a regularization approach for maximum entropy IRL. After having presented the method, an experimental evaluation on both finite and continuous environments, showing the advantages of the presented approach.

**Summary Of The Review:**

Overall, although the paper is not free of weaknesses, I think it addresses a very important problem of IRL making a first step towards the understanding of the generalization of reward functions recovered by IRL.

---

> ### Author Response · Authors · 2021-11-23
> **Authors response to Reviewer YvDw**
>
> We thank the reviewer for the encouraging review and would like to clarify the remaining issues below:
>
> 1. Similar to IRM, we do not explicity require the knowledge of interventions. We have provided
> more details on what constitutes spurious correlations and removed the emphasis on the explicit
> construction of interventional settings for the purpose of using the method.
> 2. This is a typo - we have added the missing summation in the revised manuscript.
> 3. We have made the ticks larger in the revision.
> 4. Correct, $\mathcal{E}_{tr}$ is a set of expert trajectory datasets.

---

> > ### Comment · Reviewer_YvDw · 2021-11-28
> > **Re: Authors response to Reviewer YvDw**
> >
> > I thank the authors for the answer. My positive evaluation was justified by the novelty of the approach and by the importance of the tackled problem in the IRL setting. I understand that the paper shows some important weaknesses, as also raised by other reviewers, but some of them have been addressed in the revision.

---

### Official Review · Reviewer_33fy · 2021-11-01

**Correctness:** 2
**Technical Novelty And Significance:** 2
**Empirical Novelty And Significance:** 3
**Recommendation:** 3
**Confidence:** 4

**Main Review:**

As the paper is mainly empirical (no theoretical justification for many choices), the experimental parts should provide more insights or relevant experiments to be convincing Overall the claim that "the regularization objective for inverse reinforcement learning recover robust reward functions which avoid to learn spurious correlations present in demonstration data sets" is not clearly established from the paper. Beyond the scores over a few environments that can support the generalization claim, there isn't any clear elements in the paper to support the claim about spurious correlations.

Some other elements from the paper are unclear. For instance, the following sentence is given without reference and does not seem obvious from the existing literature : "As with many neural network approximators, the discriminator model absorbs spurious correlations and this learning effect poses a serious problem. These correlations coincide with the binary label information encoding the optimality of the expert." Can you clarify, e.g. by providing relevant literature?

Is the source code provided?

Additional remarks:
- Section 3: the state space and action space are not clearly defined. Are these discrete or continuous spaces?
- In page 4, $\mathcal L^e$ is not defined, except as "individual losses"?
- $\mathcal L_{BCE}$ in 4.3 takes different arguments than the one defined from Equation 3.



**Summary Of The Paper:**

This paper investigates a specific technique for Inverse reinforcement learning to avoid the detrimental effect of "spurious correlations in the data by the learning model". The objective is to avoid "behavioural overfitting to the expert data set". The paper uses an invariant risk minimization principle as a regularization approach for the maximum entropy inverse RL problem.

**Summary Of The Review:**

The paper does not clearly backup all the claims and some elements are unclear.

---

> ### Author Response · Authors · 2021-11-23
> **Authors response to Reviewer 33fy**
>
> We thank the reviewer for the insightful comments and provide our answer below:
>
> 1. Our main supporting evidence for this claim are indeed the scores on the out-of-distribution
> environments. We have made an effort to make the concept of spurious correlations
> more clear in the revision by considering the causal graph analysis of the transition structural
> causal model.
>
> 2. In the most general case, the input to a neural network model can contain
> features which are causally correlated with the label and features which are non-causally
> correlated. Without appropriate regularization, a parametric model with a high capacity such
> as a neural network will absorb both of these correlations to make the prediction of the label,
> in our case, the binary optimality variable.
> We have added the appropriate reference for this claim in the revised manuscript.
>
> The source code is provided.
>
> - We use both discrete and continuous space for the two different types of experiments.
> - We refer to the losses incurred on individual settings. We have revised the manuscript accordingly.
> - A generic L_{BCE} formulation does not take the fixed classifier
> $w=1.0$ into account. In our algorithm, the gradient is calculated with respect to this
> classifier, hence we explicitly write out the dependence in Algorithm 1.

---

### Official Review · Reviewer_bhYo · 2021-11-02

**Correctness:** 3
**Technical Novelty And Significance:** 3
**Empirical Novelty And Significance:** 2
**Recommendation:** 5
**Confidence:** 4

**Main Review:**

The intersection of IRL and IRM is quite novel as far as I know, and at a high level it seems like a promising approach. But this paper is let down by a glaring lack of details and some paper structuring issue, with more fundamental issues lurking beneath them.

The biggest oversight is the complete lack of an appendix. Many vital details (e.g. network architecture, environment reward structure, baseline methodology) are missing from the text, to the extent that it lowers my confidence in the rest of my critique -- I'm having to guess in order to evaluate potential experimental issues. This soon be remedied as soon as possible, and I will not vote for acceptance in its absence. Code release is appreciated, but is by no means a substitute for this; the text should be able to stand alone.

1) Paper structure issues
As too much time is spent on irrelevant details of prior work, which leaves little room for critical aspects of the approach. Everything about structural causal models seems to serve no purpose. It certainly motivated IRM, but that can be left to the IRM paper itself; I see no place where this information clarifies or motivates this current method. Similarly, half a page is devoted to Figure 1, which is standard RL as inference fare, and is only used to the extent it motivates maximum entropy IRL (which is only used in the first experiment). In contrast, the jump from Equation 5 to 6 is quite vast; this is the part of the IRM paper that deserves repeating, since it adapting this regularizer that constitutes this paper's method. Similarly, equation 11 deserves unpacking. Converting the gradient of the log-likelihood loss into the difference of feature expectations is exactly the sort of prior work that needs to be understood, rather than merely cited. It is also unclear where these features come from ("output of a neural network" is far too vague), which ties into my larger complaint about the lack of relevant detail.

2) Motivational issues
2.1) Adversarial IRL/imitation learning
The combination of relying on an adversarial approach as your backbone IRL algorithm is fundamentally incompatible with your expressed desire to learning reward functions. There is a reason GAIL and related approaches are typically framed as 'imitation learning' rather than IRL: asymptotically their reward function should converge to being constant (i.e. when the policy is so good than one can't distinguish it's state occupancy from the experts'). This renders Table 1 meaningless, as better imitation should degrade the quality of the reward function. I'd suggest either switching to an non-adversarial IRL approach or dropping the idea of generating useful reward functions from the motivation.
2.2) I agree the IRL approaches tend to overfit when expert data is limited, but this is separable from the issue of generalizing to novel MDPs (i.e. different state transition dynamics). This paper conflates these two issues throughout; seemingly motivated by the first, the experiments only address the second. And this might be hard to fix. Your theory states that all experts are equally optimal, but under maximum entropy RL/RL-as-inference this would make the experts identical if the state transition dynamics of the MDP are held fixed (this is kind of the point of max Ent IRL).

3) Experimental Issues
3.1) For the non-gridworld experiments it is unclear if the baseline method is just your method without the regularization or some completely different max Ent IRL approach. I'm hoping it is the former (if not this should be included), but the text the latter and I don't understand what would motivate this.
3.2) Figure 3 results don't appear to be significant. It is also disconcerting that novel environments are being introduced with the literature is fully of similar environments with published results you could compare against.
3.3) Figure 4 is very confusing. If the experts are optimal for completely different reward functions, what should be expect imitating their aggregate policy to yield? And what is the y-axis (reward function) here?



**Summary Of The Paper:**

The authors' propose a regularization loss for inverse reinforcement learning (IRL) based on invariant risk minimisation (IRM). They argue that existing IRL methods suffer from overfitting due to the sparsity of expert trajectories, and that this is analogous to the dataset level overfitting tackled by IRM. This approach is validated by comparing against unregularized and L2 regularized maximum entropy IRL on a grid-world, and is used in a scalable adversarial IRL algorithm when dealing with more complex environments. They close by evaluating the reward function learnt by this regularized adversarial IRL in terms of its utility in training a new policy from scratch.

**Summary Of The Review:**

An interesting direction let down by poor paper structuring, a lack of detail, and a mismatch between motivation and algorithmic implementation. I highly suggest adding an appendix before addressing the other issues, as it is highly likely I've made some incorrect inferences about these missing details.

---

> ### Author Response · Authors · 2021-11-23
> **Authors comment to Reviewer bhYo**
>
> We thank the reviewer for the detailed comments and would like to provide our response regarding the weaknesses of the
> proposed method below:
>
> We have added an Appendix in our revision which will hopefully provide insights to some of the issues raised.
>
> 1. In the rest of our revision, we have moved the SCM discussion to the appendix and have provided
> the gradient derivation of equation (11) as well as network architecture and experimental
> details. We have further added a more precise description of spurious correlations in
> section 4.1 based on the causal graph analysis of the transition model.
>
> 2. (i) The AIRL discriminator [1] is explicitly formulated as the reward and
> shaping term for the purpose of extracting a reward function that can be reused in a transfer setting.
> Mathematically, one can find g and h that would yield 0.5 as the output of the disciminator in eq. 2 but still be informative as a reward function. Furthermore, it has been shown that the objective of AIRL
> matches that of solving a maximum causal entropy IRL problem ([1] Appendix A)
> so our adversarial setting also corresponds to maximum entropy IRL.
>
> (ii) Overfitting to expert data vs MDP generalization:
> - We assume that interventions on the expert trajectory SCM are caused
> by expert policy noise or dynamics changes. The sparsity of demonstrations on
> individual settings will cause the model to overfit to these, making use of any spurious correlations present.
> We leverage the fact that two experts can only simultaneously be optimal if a reward representation is invariant with respect to the individual expert variations.
>
> We make the assumption that a variation in MDP dynamics is a possible cause
> for a specific expert behaviour. This is what we refer to as an intervention on the
> dynamics. This could correspond to expert demonstrations being obtained on different
> patients in a surgical setting.
>
> The proposed method could also be employed to disentangle rewards from dynamics but
> we have not directly pursued this particular angle.
>
>
> 3.1. We apologize for the lack of clarity. The baseline method is AIRL without the proposed regularization penalty.
> 3.2. (i) We believe our method shows noticeable improvement for the Ant and HalfCheetah environments
> in Figure 3. The results for the Reacher environment are presented as a negative result where
> the chosen physical intervention to generate the expert demonstrations does not allow for a
> significant variation for the method to work correctly.
> (ii) We use commonly available PyBullet environments.
> The "Custom" label refers to the option of passing additional arguments such as
> physical parameter modifications in the form of .xml files.
> Furthermore, our evaluation settings are sampled using parameters out-of-distribution
> w.r.t. to the training settings. This particular experimental modality is
> not extensively studied in the standard benchmarks.
>
> 3.3 Y is the y-coordinate of the goal position in the plane where the robot moves.
> Based on the demonstrations obtained for the three different directions, one plausible
> solution would be a reward representation for which only the progress along the x-axis would
> be evaluated positively.
>
> [1] Fu et. al Learning Robust Rewards with Adverserial Inverse Reinforcement Learning

---

> > ### Comment · Reviewer_bhYo · 2021-11-29
> > **Thank you for the changes and clarifications.**
> >
> > Apologies for my issue 2.1; I had completely forgotten that a key difference between GAIL and AIRL papers was that the latter actually aimed to learn reward functions, not just imitate -- this is no longer a concern.
> >
> > The appendix and additional experimental details are greatly appreciated as well.
> >
> > That said, in my opinion the experimental shortcomings are still an issue.
> >
> > >We believe our method shows noticeable improvement for the Ant and HalfCheetah environments in Figure 3. The results for the Reacher environment are presented as a negative result where the chosen physical intervention to generate the expert demonstrations does not allow for a significant variation for the method to work correctly.
> >
> > The only evidence given for the Reacher experiment not providing significant variation is that it was a negative result, which is a circular argument.
> >
> > >our evaluation settings are sampled using parameters out-of-distribution w.r.t. to the training settings. This particular experimental modality is not extensively studied in the standard benchmarks.
> >
> > While I'm not aware of others testing this exact modality, the e.g. AIRL experiments (which similarly train over a distribution of dynamics parameters) could be easier adapted such that the training and test distributions were distinct. Indeed, you could smoothly vary the difference between these distribution in order to show that the degree of OOD-ness predicts the advantage of our approach. Without a connection to prior experimental results and a single particular train/test distribution mismatch being tested per environments, it is unclear how significant the advantage of your method truly is.
> >
> > Overall the paper has improved and one of my major criticisms turned out to be misplaced. But the brittleness of the experimental results prevents me from wholeheartedly endorsing this work.
> >
> > Score 3-->5

---

> > > ### Author Response · Authors · 2021-11-29
> > > **Reply to Reviewer bhYo**
> > >
> > > Thank you for the valuable feedback and the revision of the score.
> > >
> > > We appreciate the suggestion to vary the degree of OOD-ness to highlight the benefits
> > > of our method. In our experimental setup, we aimed to demonstrate that similarly to the IRM
> > > results ([2] Sec. 4.2) that a pair of training settings is sufficient to recover invariances, a
> > > single pair of settings (e.g. of dynamics variations) would also yield improved results in OOD settings
> > > compared to a pooled dataset of demonstrations using the baseline (MaxEnt / AIRL). We do agree that
> > > the results could have been presented in a more convincing manner on a more extensive evaluation set.
> > >
> > > We would also like to address the issue that the negative result on the Reacher environment
> > > constitutes a circular argument. In the manuscript, we do state that we attribute the lack of
> > > significant difference between the baseline and our method to the fact that varying the gravity
> > > coefficient does not have a physical impact on the Reacher setup since the plane on
> > > which the arm moves is frictionless, resulting in identical training settings for all variations of the parameter.
> > > We acknowledge however that we rely on this assumption without providing sufficient
> > > evidence or emphasis the text.
> > >
> > > [2] Arjovsky et al. Invariant Risk Minimization

---

### Official Review · Reviewer_nEw9 · 2021-11-08

**Correctness:** 4
**Technical Novelty And Significance:** 2
**Empirical Novelty And Significance:** 2
**Recommendation:** 3
**Confidence:** 4

**Main Review:**

Strengths:
* The work tackles the important problem of overfitting in inverse reinforcement learning. Drawing insights from invariant risk minimization the work proposes a well-motivated and theoretically supported objective.

* Figure 2 is convincing in showing that larger regularization strength results in better rewards with the IRM compared to the ERM objective.

* The paper was well written and had a good background work section.

Weaknesses:
* The method is greatly limited by the necessity of the interventions and knowing which demonstrations belong to which intervention sets. Inverse reinforcement learning (IRL) typically only requires a set of expert demonstration trajectories. This work requires significant additional information in the demonstration trajectories being partitioned as being generated from different "state interventions". This is not a realistic setting as the demonstrations need to be generated with special care for different state interventions and the learner has ground truth knowledge of these different settings.

* Insufficient comparison with regularized baselines. Recent work [4], has demonstrated that spectral normalization and mixup normalization can greatly improve the generalization capability of imitation learning agents. I imagine the same can be true for inverse-RL. This comparison is especially important because the proposed method is a form of regularization. The authors should compare to these other forms of regularization. The L2 normalization baseline also is missing from all experiments except the grid world experiment.


* Insufficient comparison with prior work in IRL. More recent methods than AIRL, such as f-IRL [3], have been proposed which perform better than AIRL. The authors should compare to the state of the art approaches in IRL.

* Insufficient comparison with prior work in generalization in imitation learning [1,2]. While the authors mention that [2] do not study IRL, it seems like the techniques from [1,2] can easily be extended to IRL?

* Insufficient experimental evidence. The main experiments only show results on a small handful of tasks. Furthermore, the Ant and HalfCheetah environments seem similar. I think experiments on more environments are necessary (such as a larger set of the Gym MuJoCo tasks).

* I cannot find any written details about the method architecture or training details either in the main paper or supplementary material.

* Presentation clarity lacks in some aspects. $ \omega $ is never defined and its role in $ \mathcal{L}_{\text{BCE}}(\xi, \phi, \omega ; e) $ is never clearly defined. What do the authors mean by a "dummy classifier"?


Questions / Clarifications:

* In what realistic settings might we have access to this intervention information?

* Why are methods trained for fewer steps in the goal intervention setting?

* Why is the variance of IRM much higher?

References:

[1] Xu, Danfei, and Misha Denil. "Positive-unlabeled reward learning." arXiv preprint arXiv:1911.00459 (2019).

[2] Zolna, Konrad, et al. "Task-relevant adversarial imitation learning." arXiv preprint arXiv:1910.01077 (2019).

[3] Ni, Tianwei, et al. "f-irl: Inverse reinforcement learning via state marginal matching." arXiv preprint arXiv:2011.04709 (2020).

[4] Orsini, Manu, et al. "What Matters for Adversarial Imitation Learning?." arXiv preprint arXiv:2106.00672 (2021).


**Summary Of The Paper:**

This work draws inspiration from the invariant risk minimization (IRM) framework to improve the generalization capability of inverse reinforcement learning (IRL) methods. The paper's contributions are deriving a new IRL objective from IRM. This new algorithm includes a penalty regularization term to the standard objective which helps the agent from overfitting to the peculiarities of the expert demonstrations.

**Summary Of The Review:**

The paper has several major issues from the large assumption of interventions in the expert dataset, to insufficient comparisons to baselines, and insufficient experimental evidence.

---

> ### Author Response · Authors · 2021-11-23
> **Authors response to Reviewer nEw9**
>
> R1: We thank the reviewer for the insightful comments and would like to address the issues below:
>
> - We would argue that the limitation in terms of necessity of intervention knowledge is not as significant as it might seem.
> The knowledge of the expert identity and consequent partitioning of the datasets into multiple
> experts is quite a common scenario e.g. demonstrations gathered from surgeons performing a procedure on different patients. In order to apply the IRM regularization method, we merely require the assumption that the expert trajectories stem from sufficiently different interventional settings. We have revised the document to emphasize that we do not explicitly require to perform an intervention on the data to construct these settings.
>
> - We are utilizing the L2 regularization with 1e-4 as the weight decay parameter in our adversarial learning experiments.
> We are also currently running the spectral norm regularization experiments to compare to our method.
>
> - The work in [2] focuses on visual features as well as imitation learning, hence we have not considered its extension
> to our setting. We had not been aware of the method presented in [1].
>
> - We agree with the reviewer that our method has only been evaluated on a limited number of domains.
> We are aiming to add more tasks to the final revision.
>
> - We have added the method architecture and training details to the appendix in the revised document.
> - The dummy classifier is a fixed scalar classifier $w=1.0$. We have revised the notation accordingly in the text.
>
> Questions:
> - One can realistically assume a scenario where experts are recorded under
> variation of dynamics. For instance, in a surgical scenario, the experts might
> perform a procedure on different patients with variations in anatomy. Our method
> can also be applied in absence of the knowledge about which specific interventions
> were made.
> - The increase in variance could be attributed to the weakening of the discriminator
> due to regularization. The reward presented to the policy during adversarial training
> will have a higher variance.

---

### Decision · Program_Chairs · 2022-01-20

**Decision:**

Reject

**Comment:**

This work addresses the issue of learning reward functions that overfit less/are invariant to irrelevant features of expert demonstrations.
The proposed algorithm builds on top of adversarial imitation learning (AIRL) and proposes to include a regularization principle that is based on invariant risk minimization. The proposed algorithm is evaluated both in grid worlds as well as continuous control tasks. Both zero-shot policy transfer, as well as transfer of the reward function to learn out-of-distribution tasks from scratch.

**Strenghts**
This work is well motivated and addresses an important problem
The proposed method is well motivated, and provides theoretical foundations

**Weaknesses**
The manuscript had many missing details/no appendix
only one baseline is provided, while many relevant IRL algorithms exist
The evaluation is very limited in actually evaluating the invariance properties of the learned reward function
poor alignment between how the proposed algorithm is motivated (learning invariant reward functions), and on what most of the experimental evaluation is focussed (zero-shot transfer of policy)**(more details on this below).


**Rebuttal**
the authors have updated the manuscript to include an appendix and were able to address most structural issues and provided many of the missing details.
No additional baselines were provided, and the experimental evaluation remains limited/poorly aligned with the initial motivation

**Summary**
This manuscript addresses an important problem and proposes a promising algorithm. My major remaining concern is the  experimental evaluation that seems not well aligned with the main contribution of this paper. As the authors state in their rebuttal the main supporting evidence for their claim is provided in Section 5.3, with only one set of experiments on using the reward function to learn policies on OOO tasks and very little analysis (< quarter of a page). While the majority of the evaluation (Section 5.2) is focussed on zero-shot transfer of the learned policy (which is trained during the IRL training phase). These zero-shot transfer experiments are not motivated in the context of the "learning invariant reward functions", so it's unclear what these results show. If these results are still relevant in showing that the proposed algorithm learns "invariant rewards", then this needs to be explained. Furthermore, more baselines would have been required (e.g algorithms that are focussed on learning a good policy by learning a "pseudo"-reward - such as GAIL).
Because of this, my recommendation is that this manuscript is not quite ready yet for publication.